# Does Radiomic Segmentation Complexity Influence Foundation Model Performance? A Case Study with SAM-Med3D

**F. Javier Gil-Terrón**[1]                                          FRAGILRO@UPV.ES

**Carles Lopez-Mateu**[1]                                          CLOPMAT@UPV.ES

**María Gómez-Mahiques**[1]                                        MAGOMA2@UPV.ES

**Víctor Montosa-i-Mico**[1]                                        VICMONM5@UPV.ES

**Juan Miguel García-Gómez**[1]                               JUANMIG@IBIME.UPV.ES

**Elies Fuster-Garcia**[1]                                          ELFUSGAR@UPV.ES

[1]*Biomedical Data Science Laboratory, Instituto Universitario de Tecnologías de la Información y Comunicaciones, Universitat Politècnica de València, 46022 Valencia, Spain*

**Editors:** Accepted for publication at MIDL 2025

## Abstract

The Segment Anything Model (SAM) has expanded the application of foundation models in medical image segmentation. However, performance can vary significantly depending on the complexity of the segmentation task. This study examines how segmentation complexity, characterized through radiomic features, impacts the performance of SAM-Med3D in 3D medical imaging tasks. It explores the relationship between segmentation complexity and model performance using five public datasets: MSD-Vessel, MSD-Colon, EPISURG, SPI-DER, and PENGWIN. The analysis computed Intersection over Union (IoU) and Mean Surface Distance (MSD). Our results revealed that radiomic features such as mesh volume, sphericity, surface/volume ratio, and texture difference inside and outside the ROI significantly correlate with segmentation performance. Higher mesh volumes and lower surface/volume ratios were associated with better performance, suggesting that more compact and larger structures are segmented more accurately. These findings underscore the relevance of assessing the influence of segmentation complexity in medical imaging, as captured through radiomic features. This analysis provides valuable insights into the applicability of generalist models to specific tasks, based on the radiomic characteristics of the data.

**Keywords:** Medical Image Segmentation, Foundation Model, Deep Learning, Radiomics.

## 1. Introduction

Recently, the release of the Segment Anything Model (SAM) (Kirillov et al., 2023) has increased the application of foundation models to image segmentation tasks, including those in the medical field (Azad et al., 2023; Ma et al., 2023; Zhang et al., 2024, 2023a). Since SAM was originally designed for 2D images, numerous approaches have been developed to extend SAM's capabilities to handle the complexity of 3D data, creating new foundation models specifically for 3D medical imaging (Wu et al., 2023; Chen et al., 2023; Li et al., 2023a; Bui et al., 2023; Zhang et al., 2023b; Li et al., 2023b).

While most evaluation studies focus primarily on the overall performance, recent research has also examined how the region of interest (ROI) characteristics influence segmentation outcomes. Prior work by (Mazurowski et al., 2023) analyzed SAM's segmentation

performance demonstrating that well-defined structures tend to yield higher segmentation accuracy compared to more complex ROIs. Despite these findings, a detailed investigation into the role of radiomic complexity in foundation model performance remains limited.

This work examines how ROI complexity, quantified through radiomic features, influences the performance of SAM-Med3D (Wang et al., 2023). The analysis covered different types of medical images, providing insights into how these features correlate with model performance. Equations relevant to this analysis are provided in the appendix for reference.

## 2. Materials

This study utilized five sets of medical imaging tasks. **EPISURG** (Pérez-García et al., 2020): 133 T1-weighted MRIs of epileptic patients with post-surgery lesion segmentations, **PENGWIN** (Liu et al., 2023): 100 3D CT samples from the Pelvic Fracture Segmentation challenge, **SPIDER** (van der Graaf et al., 2024): 196 T2-weighted lumbar spine MRIs with segmentations of vertebrae discs and spinal canal, Medical Segmentation Decathlon: Task 10 Colon **MSD-Colon** (Antonelli et al., 2022): 126 CT scans for colon cancer segmentation, and Medical Segmentation Decathlon: Task 08 Hepatic Vessels **MSD-Vessel** (Antonelli et al., 2022): 303 CT scans for segmenting tubular structures near heterogeneous tumors.

## 3. Methods

For this work, the SAM-Med3D (Wang et al., 2023) foundation model was used. Images were interpolated to 1.5mm spacing, z-normalized (excluding background), and padded to a minimum size of 128 in each dimension. Evaluation was performed using a sliding window, with input prompts generated by the original SAM-Med3D based on 11 random ROI clicks.

For evaluation, the Intersection over Union (IoU) (Equation (1)) and Mean Surface distance (MSD) (Equation (2)) were computed. The relationship between these metrics and radiomic features was analyzed through regression analyses, with each metric regressed against each feature to identify those most associated with segmentation performance.

Analyzed radiomic features included: **Mesh Volume** (C), ROI volume from the segmentation mesh, providing precise geometric measurement. **Shape Sphericity** (D), quantifies how sphere-like the segmentation is, ranging from 0 to 1, with 1 indicating a perfect sphere. It reflects shape regularity independent of scale and orientation. **Surface/Volume Ratio A/V**, measures surface area relative to volume, with lower values indicating compact shapes and higher values reflecting surface complexity or irregularity. **Texture** (E), used GLCM and GLRLM to extract feature vectors from regions 10 voxels inside and outside the ROI boundary. Differences between these vectors captured texture contrast and boundary complexity, reflecting interactions between the lesion and surrounding tissue.

## 4. Results

Table 1 shows the results of separate regression analyses, where each performance metric (IoU and MSD) was regressed individually against each radiomic feature. The table reports the corresponding slopes (representing the change in the metric per unit increase in the feature), along with $R^2$ and p-values. Mesh volume was positively associated with IoU,

showing an increase of 3.55e-7 per unit (p-value = 1.14e-10), and negatively associated with MSD, decreasing by 9.30e-6 per unit (p-value = 4.78e-6). Surface/volume ratio showed a strong negative correlation with IoU, decreasing by 0.5434 per unit (p-value = 7.61e-15). Sphericity was significantly associated with higher MSD, increasing by 15.66 per unit (p-value = 5.39e-5). Texture difference at the ROI boundary was also linked to a significant increase in MSD of 1.0831 per unit (p-value = 0.0234).

Table 1: Slope (change in metric per unit increase in the radiomic feature), $R^2$, and p-values from the regression analysis of the SAM-Med3D model for both IoU and MSD. Overall statistically significant p-values ($\leq 0.05$) are shown in bold.

| Feature | IoU Slope | $R^2$ | p-value | MSD Slope | $R^2$ | p-value |
|---|---|---|---|---|---|---|
| Mesh volume | 3.55e-7 | 0.3848 | **1.14e-10** | -9.30e-6 | 0.2170 | **4.78e-6** |
| Sphericity | 0.1107 | 0.0105 | 0.3411 | 15.6604 | 0.1736 | **5.39e-5** |
| Surface/Volume | -0.5434 | 0.5066 | **7.61e-15** | 4.8217 | 0.0328 | 0.0910 |
| Texture difference | -0.0070 | 0.0030 | 0.6113 | 1.0831 | 0.0528 | **0.0234** |

## 5. Discussion and Conclusions

Regression analysis revealed that the model performance was strongly influenced by radiomic features, with mesh volume showing significant correlations with both IoU and MSD, aligning with prior studies (Lei et al., 2023; Gong et al., 2024). Surface/volume ratio negatively correlated with IoU, indicating difficulties in segmenting irregular shapes, while sphericity and texture difference primarily affected MSD, reflecting boundary-level challenges. These results suggest that segmentation difficulty is driven not only by size but also by geometric and textural complexity—tasks with low sphericity, high surface/volume ratio, and greater texture variation tend to be more challenging for the model.

These findings highlight the importance of considering segmentation complexity, measured through radiomic analysis, when evaluating task difficulty, as it directly impacts model performance. Further research is needed to determine how well these insights generalize across various medical imaging applications and segmentation tasks. Future work should also explore adaptive learning strategies, architectural improvements, and task-specific training methods to enhance performance in more complex segmentation scenarios.

This study demonstrates that segmentation complexity, characterized through radiomic features, conditions the performance of foundation models like SAM-Med3D in 3D medical imaging tasks. Understanding these relationships could enable the anticipation of model performance on unseen datasets, opening the door to predictive strategies that estimate how well a foundation model will perform based on the radiomic properties of the task.

## Acknowledgments

Work funded by Grant PID2021-127110OA-I00 (PROGRESS), MCIN/AEI/ 10.13039/501100011033, ERDF SINUÉ (INNEST/2022/87 – Agencia Valenciana de la Innovación), ALBATROSS (PID2019-104978RB-I00), Programa INVESTIGO (INVEST/2022/298) and Grant CIAICO/2022/064 (Lalaby-Glio) by Generalitat Valenciana.

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

## Appendix A. Intersection over Union (IoU) or Jaccard index

The IoU of two elements, A and B, is defined as the ratio of the intersection of their sets (A ∩ B) to their union (A ∪ B). In this equation, TP represents true positives (areas correctly identified as belonging to the target), FP represents false positives (areas incorrectly identified as belonging to the target), and FN represents false negatives (areas belonging to the target but missed by the prediction).

$$IoU(A, B) = \frac{|A \cap B|}{|A \cup B|} = \frac{TP}{(TP + FP + FN)} \tag{1}$$

## Appendix B. Mean Surface distance (MSD)

The MSD between two elements, A and B, is defined as the average of the shortest distances from the surface of A to the surface of B and vice versa. In this equation, $S_A$ and $S_B$ represent the surfaces of the respective segmentations, while $d(a, S_B)$ and $d(b, S_A)$ denote the shortest Euclidean distances from a point on one surface to the other. This metric provides a measure of the spatial discrepancy between segmentation boundaries, with lower values indicating better alignment.

$$MSD(A, B) = \frac{1}{|S_A| + |S_B|} \left( \sum_{a \in S_A} \min_{b \in S_B} ||a - b|| + \sum_{b \in S_B} \min_{a \in S_A} ||b - a|| \right) \tag{2}$$

## Appendix C. Mesh Volume

The volume of the ROI $V$ is calculated from the triangle mesh of the ROI. For each face $i$ in the mesh, defined by points $a_i$, $b_i$ and $c_i$, the (signed) volume $V_f$ of the tetrahedron defined by that face and the origin of the image ($O$) is calculated. (Equation (3)) The sign of the volume is determined by the sign of the normal, which must be consistently defined as either facing outward or inward of the ROI.

Then taking the sum of all $V_i$, the total volume of the ROI is obtained (Equation (4))

$$V_i = \frac{O_{a_i} \cdot (O_{b_i} \times O_{c_i})}{6} \tag{3}$$

$$V = \sum_{i=1}^{N_f} V_i \tag{4}$$

## Appendix D. Shape sphericity

$$sphericity = \frac{\sqrt[3]{36\pi V^2}}{A} \tag{5}$$

## Appendix E. Texture Vector

The texture vector in this study is composed of features derived from two key methods: Gray Level Co-occurrence Matrix (GLCM) and Gray Level Run Length Matrix (GLRLM). GLCM features include Contrast, Correlation, Joint Energy, Maximal Correlation Coefficient, and Joint Entropy, while GLRLM features consist of Short Run Emphasis, Long Run Emphasis, Gray Level Non-Uniformity, and Run Percentage. These features provide a comprehensive representation of texture patterns within and around the segmentation regions.

Contrast (Equation (6)) is a measure of the local intensity variation, favoring values away from the diagonal $(i = j)$. A larger value correlates with a greater disparity in intensity values among neighboring voxels.

$$contrast = \sum_{i=1}^{N_g} \sum_{j=1}^{N_g} (i-j)^2 p(i,j) \tag{6}$$

Correlation (Equation (7)) is a value between 0 (uncorrelated) and 1 (perfectly correlated) showing the linear dependency of gray level values to their respective voxels in the GLCM.

$$correlation = \frac{\sum_{i=1}^{N_g} \sum_{j=1}^{N_g} p(i,j)ij - \mu_x\mu_y}{\sigma_x(i)\sigma_y(j)} \tag{7}$$

Energy (Equation (8)) is a measure of homogeneous patterns in the image. A greater Energy implies that there are more instances of intensity value pairs in the image that neighbor each other at higher frequencies.

$$\text{Joint Energy} = \sum_{i=1}^{N_g} \sum_{j=1}^{N_g} (p(i,j))^2 \tag{8}$$

The Maximal Correlation Coefficient (Equation (9)) is a measure of complexity of the texture and $0 \leq MCC \leq 1$. In case of a flat region, each GLCM matrix has shape (1, 1), resulting in just 1 eigenvalue. In this case, an arbitrary value of 1 is returned.

$$MCC = \sqrt{\text{second largest eigenvalue of Q}} \tag{9}$$

$$Q(i,j) = \sum_{K=0}^{N_g} \frac{p(i,k)p(j,k)}{p_x(i)p_y(k)} \tag{10}$$

Joint entropy (Equation (11)) is a measure of the randomness/variability in neighborhood intensity values.

$$\text{Joint Entropy} = -\sum_{i=1}^{N_g} \sum_{j=1}^{N_g} (p(i,j))log_2(p(i,j) + \epsilon) \tag{11}$$

Short Run Emphasis (SRE) (Equation (12)) is a measure of the distribution of short run lengths, with a greater value indicative of shorter run lengths and more fine textural textures.

$$SRE = \frac{\sum_{i=1}^{N_g} \sum_{j=1}^{N_g} \frac{P(i,j|\theta)}{j^2}}{N_r(\theta)} \tag{12}$$

Long Run Emphasis (LRE) (Equation (13)) is a measure of the distribution of long run lengths, with a greater value indicative of longer run lengths and more coarse structural textures.

$$LRE = \frac{\sum_{i=1}^{N_g} \sum_{j=1}^{N_g} P(i,j|\theta)j^2}{N_r(\theta)} \tag{13}$$

Gray Level Non-Uniformity (GLN) (Equation (14)) measures the similarity of gray-level intensity values in the image, where a lower GLN value correlates with a greater similarity in intensity values

$$GLN = \frac{\sum_{i=1}^{N_g} (\sum_{j=1}^{N_g} P(i,j|\theta))^2}{N_r(\theta)} \tag{14}$$

Run Percentage (RP) (Equation (15)) measures the coarseness of the texture by taking the ratio of number of runs and number of voxels in the ROI. Values are in range $\frac{1}{N_p} \leq RP \leq 1$, with higher values indicating a larger portion of the ROI consists of short runs (indicates a more fine texture).

$$RP = \frac{N_r(\theta)}{N_p} \tag{15}$$

