# OpenReview forum: "Does Radiomic Segmentation Complexity Influence Foundation Model Performance? A Case Study with SAM-Med3D"
_MIDL.io/2025/Short_Papers — MIDL 2025 - Short Papers_

### Official Review · Reviewer_eogW · 2025-04-25

**Rating:** 3
**Confidence:** 4

**Summary:**

The manuscript investigates correlation between certain radiomics features and MedSAM3D segmetnation ability.

**Strengths:**

+ It is, in principle, interesting to better understand udner what circumstances SamMED3D performs well and when it does not

+ use of several public benchmarks

**Weaknesses:**

- The choice of radiomics features seems odd; only texture (and also only partially) can be evaluated without access to ground truth (volume, surface, and ratios cannot be assessed and even texture is calculated within and outside requireing definition of a boundary). This makes the idea much less powerful, because one cannot use radiomics features now to forecast likely sammed failure. It would be much more interesting if one was able to predict that SemMED is likely or unlikely to segment correctly given radiomics features in a certain area.

---

### Decision · Program_Chairs · 2025-05-01

Accept